# Implementing the compassion intervention, a model for integrated care for people with advanced dementia towards the end of life in nursing homes: a naturalistic feasibility study

Kirsten J Moore,[1] Bridget Candy,[1] Sarah Davis,[1] Anna Gola,[1] Jane Harrington,[1] Nuriye Kupeli,[1] Victoria Vickerstaff,[1] Michael King,[2] Gerard Leavey,[3] Irwin Nazareth,[4] Rumana Z Omar,[5] Louise Jones,[1] Elizabeth L Sampson[1]

► Prepublication history and additional material are available online. To view these files please visit the journal online (http://dx.doi.org/10.1136/bmjopen-2016-015515).

[1]Marie Curie Palliative Care Research Department, Division of Psychiatry, University College London, London, UK
[2]Division of Psychiatry, University College London, London, UK
[3]Bamford Centre for Mental Health & Wellbeing, University of Ulster, Derry Londonderry, UK
[4]Department of Primary Care and Population Health, University College London, UK
[5]Department of Statistical Science, University College London, London, UK

**Correspondence to**
Dr. Kirsten J Moore; kirsten.moore@ucl.ac.uk

## ABSTRACT

**Background** Many people with dementia die in nursing homes, but quality of care may be suboptimal. We developed the theory-driven 'Compassion Intervention' to enhance end-of-life care in advanced dementia.

**Objectives** To (1) understand how the Intervention operated in nursing homes in different health economies; (2) collect preliminary outcome data and costs of an interdisciplinary care leader (ICL) to facilitate the Intervention; (3) check the Intervention caused no harm.

**Design** A naturalistic feasibility study of Intervention implementation for 6 months.

**Settings** Two nursing homes in northern London, UK.

**Participants** Thirty residents with advanced dementia were assessed of whom nine were recruited for data collection; four of these residents' family members were interviewed. Twenty-eight nursing home and external healthcare professionals participated in interviews at 7 (n=19), 11 (n=19) and 15 months (n=10).

**Intervention** An ICL led two core Intervention components: (1) integrated, interdisciplinary assessment and care; (2) education and support for paid and family carers.

**Data collected** Process and outcome data were collected. Symptoms were recorded monthly for recruited residents. Semistructured interviews were conducted at 7, 11 and 15 months with nursing home staff and external healthcare professionals and at 7 months with family carers. ICL hours were costed using Department of Health and Health Education England tariffs.

**Results** Contextual differences were identified between sites: nursing home 2 had lower involvement with external healthcare services. Core components were implemented at both sites but multidisciplinary meetings were only established in nursing home 1. The Intervention prompted improvements in advance care planning, pain management and person-centred care; we observed no harm. Six-month ICL costs were £18 255.

**Conclusions** Implementation was feasible to differing degrees across sites, dependent on context. Our data inform future testing to identify the Intervention's effectiveness in improving end-of-life care in advanced dementia.

### Strengths and limitations of this study

► This feasibility study informs future testing of the Compassion Intervention to identify its effectiveness in improving end-of-life care for residents with advanced dementia and their families.

► We followed principles of dynamic sustainability, recognising that implementing protocols in real-life settings requires adaptations, and that rigid adherence to guidelines tested in controlled settings may not be suitable or effective in broader contexts.

► We structured our approach using the five phases of implementation described in the literature on whole systems change in healthcare including orientation, insight, acceptance, change and maintenance.

► Recognising the importance of context on implementation, we report on four levels of nursing home context: political and economic; organisational; social; and individual professionals.

► As an exploratory study the sample size was small and we did not aim to detect differences or calculate a sample size for future studies.

**Trial registration** ClinicalTrials.gov:NCT02840318: Results

## INTRODUCTION

Dementia is the fourth most common cause of death in high-income countries[1] where most people with dementia die in long-term care institutions including nursing homes (NHs).[2–4] The European Association for Palliative Care (EAPC) defines good care for people with dementia approaching death as person-centred, involving shared decision-making with the person with dementia and family members.[5] This may require an integrated approach[6] and a central care coordinator.[5] UK policy states that care is integrated when 'people benefit

from care that is person-centred and co-ordinated within healthcare settings, across mental and physical health and across health and social care. For care to be integrated, organisations and care professionals need to bring together all of the different elements of care that a person needs.'[7]

Currently, barriers to integrated care remain.[8] Many NH residents experience burdensome interventions and distressing symptoms during the last months of life.[9] Recent data show higher emergency admissions among older people residing in NHs,[10] indicating persistent gaps in healthcare planning.

Providing good end of life (EOL) dementia care is complex, prognosis is unpredictable[11] and managing symptoms is difficult when communication is compromised. The need for a complex intervention is reflected in the EAPC's 57 recommendations for optimal EOL dementia care.[5] However, interventional research on providing EOL care in dementia is scant[12] and lacks a theoretical basis.[13]

Establishing a complex intervention begins with development based on the available evidence and theories, testing its acceptability and feasibility in practice, evaluation via larger trials through to wider dissemination into practice.[14] Practice change theories highlight the challenge of incorporating interventions into practice and the need to consider the effect of context at societal, organisational and individual levels.[15]

Few other interventions have been specifically developed to improve EOL care in advanced dementia. In the USA, an interdisciplinary approach towards individualised care plans for residents with advanced dementia achieved this by creating new hospice units within the long-term care setting rather than attempting to change NH practice.[16] A protocol for an Australian trial describes a study to be conducted that aims to promote family case conferencing through training NH nurses to work as palliative care coordinators and involving family, NH staff and healthcare professionals in case conferences for residents with advanced dementia.[17] In the UK, the Gold Standards Framework in Care Homes (GSFCH) and the ABC EOL Education Programme promote a palliative approach within care homes (including NHs), although not specifically for residents with dementia.[18 19] Further studies of the GSFCH have found that most care homes fail to pass the accreditation standard and that high facilitation with additional action learning sessions increased accreditation rates from 7% to 83%.[18] This suggests that education programmes alone are unlikely to change resistant norms and practices.[20]

### The Compassion Intervention
Within a 3-year research programme funded by Marie Curie Care (National Institute for Health Research, Primary Care Research Network Refs. 12621; 12623),[21] we used the RAND/University of California, Los Angeles (UCLA) Appropriateness Method[22] to achieve national consensus on the components of Compassion ('the Intervention'), a complex model of EOL care for people with advanced dementia. The development of the Intervention has been reported,[6] is based on theories of multilevel and whole

systems change,[15 23] and is described in detail in a manual (available on the UCL Marie Curie Palliative Care Research Department website).

The Intervention is aimed at people aged 65 years and over who have advanced dementia using criteria based on an existing model of UK best practice:[24] (1) memory problems indicating a diagnosis of dementia according to the fourth Diagnostic and Statistical Manual of Mental Disorders; (2) Functional Assessment Staging Scale grade 6a (difficulty putting on clothing) through to 7f (unable to hold head up);[25] (3) comorbidities or unmanaged symptoms such as agitation, recurrent infections, pain and pressure ulcers.

There are two core components: facilitation of an integrated, multidisciplinary approach to assessment, treatment and care; and education, training and support for formal and informal carers (table 1). The Intervention is coordinated by an interdisciplinary care leader (ICL) who scopes local practice and identifies key personnel to support EOL care. Scoping ensures the Intervention complements, rather than duplicates, existing local processes. The ICL establishes and coordinates key activities to address the two core components of the Intervention (table 1). Activities to facilitate component 1 include: (1) person-centred assessment of residents, focusing on their physical, psychological, emotional and social needs, (2) meetings of the core care team and the wider multidisciplinary care teams. Activities to facilitate component 2 include: (3) staff training sessions, education and support for NH staff and family carers. The ICL role requires a broad range of skills including clinical experience in care of frail older people and those with dementia, particularly towards EOL, ability to educate staff and talk empathically with family carers, and sensitivity to identify and minimise poor care practices. Skills may be drawn from the fields of nursing, social work or a profession allied to medicine.

The Intervention has similar components to existing EOL programmes in care homes such as education provision[18 19] and multidisciplinary input.[17] The key distinguishing feature of the Intervention is the role of the independent ICL who works solely with two NHs to provide mentoring, role modelling, advice and training and who can develop relationships with NH staff, external healthcare professionals, residents and family carers and develop an in-depth understanding of the organisational culture underpinning practice and impacting on practice change initiatives.

### Aim
We aimed to (1) understand how the Intervention operated in two NHs in different health and social care settings; (2) collect preliminary outcome data and estimate the cost of employing an ICL to inform further evaluative studies; (3) check that the Intervention caused no physical or psychological harm to residents or their family carers.

**Table 1** Key activities of the Compassion Intervention

| Component and activity | Purpose | Who is involved | Content |
|---|---|---|---|
| 1: facilitation of an integrated, multidisciplinary approach to assessment, treatment and care: a) Individual holistic resident assessment | To identify symptoms, areas of current unmet need, anticipated future needs and corresponding actions and goals. | The ICL assesses eligible residents in conjunction with NH nurses and healthcare assistants. The process involves liaison with the resident and family about their perceived needs, issues and expectations regarding EOL care. The assessment involves observations and if possible, discussions with the resident. The assessment template focuses on observational measures to identify whether the resident is showing signs of comfort, discomfort, distress and/or pain. | Assessment template:<br>•Dementia diagnosis and progression (Functional Assessment Staging Scale)<br>•Significant other medical conditions<br>•Life history, interests<br>•Important goals for care and well-being<br>•Needs or restrictions related to faith and/or culture<br>•EOL wishes (Did the resident document preferences when they had capacity? Are family carer preferences documented? Are resuscitation status and preferred place of death documented and reviewed?<br>•Current medication (and recent changes)<br>•Level of meaningful communication and understanding<br>•Presence of pain or discomfort (Pain Assessment in Advanced Dementia Scale)<br>•Behavioural symptoms and sleep disturbance<br>•Psychological well-being, mood, anxiety or depression (Cornell Scale for Depression in Dementia)<br>•Mobility, falls risk, sitting balance and posture, contractures/tone<br>•Skin conditions, pressure sore risk (Waterlow Score)<br>•Continence, constipation/bowel problems, UTIs<br>•Eating and swallowing, oral care, weight loss, nutritional status<br>•Other problems—chest infections, breathlessness, fits, blackouts<br>•Recent change in condition<br>•Summary of unmet needs and anticipated/ future needs<br>•Action plan and goals. |
| 1: facilitation of an integrated, multidisciplinary approach to assessment, treatment and care: b) Weekly core meetings | To review, agree on and enact (including referrals), the individual holistic resident assessments. | The core team includes those responsible for medical, nursing and social needs of residents and may include: the clinician responsible for the resident's medical needs (GP, geriatrician or old age psychiatrist), NH nursing staff responsible for the resident's needs, and the ICL. | Review of individual assessments including developing an action plan to address areas of unmet need, discussion of anticipated needs, an escalation plan for the most likely 'what ifs', review of medications and prescribing 'just in case' medications if appropriate and review of EOL wishes and resuscitation status to ensure these are clearly documented. A review date and whether the resident's needs require discussion with the wider team will be decided. |

Continued

**Table 1** Continued

| Component and activity | Purpose | Who is involved | Content |
|---|---|---|---|
| 1: facilitation of an integrated, multidisciplinary approach to assessment, treatment and care: c) Monthly wider team meetings | To discuss (in person or via teleconference), complex cases and review care plans, consider significant events, critical incident analysis. | The wider team will consist of the core team plus any local health and social care professionals and specialist services involved in the care of people with advanced dementia. This is likely to include general practice, care of the elderly, old age psychiatry, palliative care, social services and community services such as district nursing, speech and language therapy, dietetics, tissue viability, physiotherapy and occupational therapy. Composition will depend on local working practices and the availability of key personnel. | The core team will present for discussion residents who have complex needs requiring specialist advice or those where actions agreed by the core team have not been successful at alleviating symptoms. The wider team will also consider learning or training needs that may become evident as a consequence of this shared working. The meetings will include discussion of critical incidents, deaths, hospital admissions, complaints or compliments, and significant events relating to the care of residents so that learning points can be identified. |
| 2: Education, training and support for formal and informal carers | To establish and address the educational needs of staff members so that they can recognise and respond effectively to the needs of people with advanced dementia and to support family carers with increased confidence. | ICL will work with the NH and wider team to identify and address education needs and will obtain agreement from the NH manager to run formal training sessions. The ICL will be supported by the wider team to undertake training and education. The target of training could include staff and family carers. | EOL care for people with advanced dementia linking to core competencies outlined in reference 54[54] including:<br>•Communication skills with residents with advanced dementia and family carers<br>•Assessment and care planning<br>•Symptom management to maintain comfort and well-being<br>•Advance care planning<br>•Knowledge and values, to understand advanced dementia and EOL care and when to refer to specialist services. To be sensitive to the needs of family carers and to foster respect, dignity and quality care. |

EOL, end of life; GP, general practitioner; ICL, interdisciplinary care leader; NH, nursing home; UTI, urinary tract infection.

## METHOD

A naturalistic feasibility study of the Compassion Intervention. We followed the principles of dynamic sustainability, recognising that implementing protocols in real life settings requires adaptations, and that rigid adherence to guidelines tested in controlled settings may not be suitable or effective in broader contexts.[26] We structured our approach using the five phases of implementation described by Grol:[23] (1) Orientation (awareness of the need for a revised model of care; interest and involvement in the work) (2) Insight (understanding of the revised model of care; insight into existing routines of care) (3) Acceptance (positive attitudes to the possibilities of developing practice; a decision to explore change) (4) Change (actual adoption of a new care model; try-out and confirmation of value) (5) Maintenance (new practice integrated into routines; new practice embedded in the organisation).

Recognising the importance of context on implementation, we report on four levels of NH context: political and economic; organisational; social; and individual professionals.[23]

We employed a full-time ICL (KJM) with a social care background and experience of working with people with dementia in NHs. The ICL received supervision from clinicians with palliative and dementia expertise. Two NHs were invited to participate; both were involved earlier in our research programme and provided data for a longitudinal (9 months) cohort study to understand the clinical context of people with advanced dementia and their family carers.[21] NH managers identified eligible residents. We aimed to assess two residents in each NH per week (activity 1a, table 1).

Implementation occurred over 6 months at each site (see published protocol,[27] supplementary file 1 and supplementary file 2). In month 1, the ICL met with NH managers and key external healthcare professionals, introduced herself to staff and displayed study posters. The Intervention was launched in nursing home 1 (NH1) in May 2014 and nursing home 2 (NH2) in June 2014. Table 1 shows the activities led by the ICL and after 6 months the ICL ceased active engagement. To assess maintenance of activities, interviews with relevant stakeholders were conducted after the ICL withdrew at months 7, 11 and 15. Participants were recruited from May 2014 to August 2015. The nature of the intervention prevented masking but independent researchers collected individual-level resident and carer data and conducted qualitative interviews.

### Data collection
#### Scoping of existing context
The ICL interviewed each NH manager prior to launching the Intervention. Topics included: resident characteristics, staffing levels, care planning and communication processes, access to external healthcare professionals, training opportunities, dementia and palliative care, and expectations about the Intervention. This was supplemented through meetings with deputy managers and other external healthcare professionals.

#### Qualitative and quantitative process data recorded by ICL
The ICL kept a (1) reflective diary recording observations of practice, liaison with staff, family and residents, examples of improvements in care and personal responses to the role;[28] (2) a daily log of time spent on tasks related to implementation to enable estimation of costs. We assumed that staff time spent in meetings and training was consistent with usual working practice and so was not considered an additional cost; any opportunity costs incurred would have been offset by the training skills acquired.

Over 6 months at each site, the ICL collected monthly NH-wide data on the number of residents with: documented resuscitation status; a pain management plan; preferred place of death recorded; hospital admissions as possible indicators of quality of EOL care. Data on emergency phone calls and location of deaths were also collected for this purpose. Resident assessments undertaken by the ICL (Activity 1a, table 1) were part of routine care and were maintained within the NH as clinical information according to their governance polices. Findings from assessments could be reflected on in the anonymised ICL diary and used to inform other Intervention activities such as training. Formal training sessions with staff and family (Activity 2, table 1) were formally evaluated by participants.

#### NH resident and carer data
Monthly individual outcome data from participant residents who had been assessed by the ICL and their family carers were collected by researchers (NK, SD). Residents were recruited during the first 4 months of implementation to enable at least 3 months of outcome data. We used measures from our earlier cohort study for simple comparisons and to check for potential harm.[21] To describe the sample at baseline we used the Functional Assessment Staging Scale,[25] the Charlson Comorbidity Index[29] and Bedford Alzheimer Nursing Severity Scale.[30] To assess resident outcomes we used the Waterlow Scale (pressure ulcer risk),[31] Neuropsychiatric Inventory,[32] Cohen-Mansfield Agitation Inventory,[33] Pain Assessment in Advanced Dementia Scale,[34] Symptom Management at EOL in Dementia[35] and Quality of Life in Late-Stage Dementia Scale.[36] For carer outcomes we used the 22-item Zarit Burden Interview,[37] the Hospital Anxiety and Depression Scale,[38] Satisfaction with Care at EOL in Dementia Scale[35] and the Resource Utilization in Dementia Questionnaire.[39]

#### Qualitative interview data from staff and family carers
We conducted semistructured interviews with a purposive representative sample of NH staff and attending professionals at three time points (months 7, 11 and 15) after the ICL left the site. Family carers who had agreed for a resident to have monthly individual data collected were invited

for interview at month 7. Interviews were audio recorded and transcribed verbatim. We aimed to: assess participants' views of the strengths and weaknesses of the Intervention; identify whether any changes in practice were implemented due to the Intervention; and explore whether these changes were maintained after the ICL left.

## Analysis

### Qualitative analysis

Transcripts were checked against the audio recording. One researcher involved in interviewing and transcribing (NK) reread and coded all transcripts using QSR International NVivo V.10 software (2012). Framework analysis was used,[40] based on the five phases of implementation.[23] Small chunks of text were extracted and coded, summarising their content. NK categorised each piece of coded text under each of the five phases. After all coded text was categorised, codes were grouped into a smaller number of themes within each phase of implementation. Additional details about each category reported by Grol et al[23] were also used to inform the categorisation process. The revised structure was reviewed by GL to check for agreement with interpretation. This led to an additional theme being incorporated into the context section of the results. Themes were evident in both NHs, unless identified otherwise.

### Quantitative analysis

Process data are reported as total number of activities (as outlined in table 1) undertaken and total ICL hours spent on different activities. ICL hours spent on activities associated with the implementation were costed using the Department of Health and Health Education England tariffs to estimate the cost of engaging the ICL. Training evaluations and outcomes (facility-wide and individual) are reported using descriptive statistics using statistical package IBM SPSS V.22 (2013). Outcome data were used for monitoring potential harm and to examine the feasibility of collecting measures in future trials, hence a sample size calculation was not performed. For individual assessments we present outcome measures from the last available assessment using descriptive statistics. We also compare these measures with data from our earlier cohort study but have not made statistical comparisons due to an anticipated small sample size.

## Ethics approval and consent to participate

NH managers gave written consent for their site to participate, and permission for the ICL to carry out clinical assessments of eligible residents and have access to their files. None of the residents had the capacity to make an informed decision for research participation so NH managers invited their next of kin/primary contact to give agreement. If next of kin were not available, a professional consultee provided agreement according to the Mental Capacity Act (2005). Staff and family gave written informed consent prior to each interview.

## RESULTS

We begin by describing the NH context based on the experiences of the ICL, data collected during set-up and qualitative interviews. We describe how the Intervention operated in practice from experiences of the ICL and qualitative interviews. We report the extent to which the core Intervention activities (table 1) were possible. We present findings from the qualitative interviews to understand the five phases of implementation: orientation, insight, acceptance, change and maintenance.[23] Finally we present individual and NH-wide outcomes and cost data to inform future testing or commissioning of a similar intervention. Figure 1 provides a flow chart of all participants. In total 48 interviews were conducted (NH1=30; NH2=18) with 28 NH and external healthcare professionals at 7 (n=19), 11 (n=19) and 15 months (n=10). Four family carers all from NH2 were interviewed at 7 months.

### Context

Supplementary file 3 describes both NHs according to contextual levels; political and economic, organisational, social, and individual professionals.[23] While both NHs were located within the same broader political and economic contexts, they also operated within different local funding systems for healthcare and social care services (Clinical Commissioning Groups; CCGs). NH1 was located in a more socioeconomically deprived area.[41] Both NHs were located in CCGs with priorities around EOL, but only the NH1 CCG also had a priority relating to care for the 'frail and elderly'.[42 43] NH1 was located in a CCG with fewer NHs than NH2. Both NHs were part of larger private companies and both had contracts with one general practitioner (GP) surgery with the goal of having one GP oversee the medical care of all residents within the NH. Key functional differences between NH1 and NH2 related to access and involvement with external healthcare services, level of detail in care planning processes, and procedures for training for staff, all indicating greater support and development of processes in NH1. While NH1 only contained nursing beds (99 beds with 85 for older people), NH2 had three units with only two of these providing nursing care (52 beds). The third unit (25 beds) was a residential unit with visiting nurses only; residents from here were not assessed during the Intervention.

During implementation and through in-depth qualitative interviews, we found that the context of both NHs was characterised by poor knowledge in dementia and EOL care. Training needs were identified in: pain management, clinical observation and needs assessment, communication with family and residents, advance care planning, person-centred care, psychological aspects of dementia and transition planning. For example, concerns were raised by NH nurses and external healthcare professionals about the confidence of NH nurses having EOL conversations with family:

'…often these conversations are quite difficult to conduct and it needs time and it needs some background

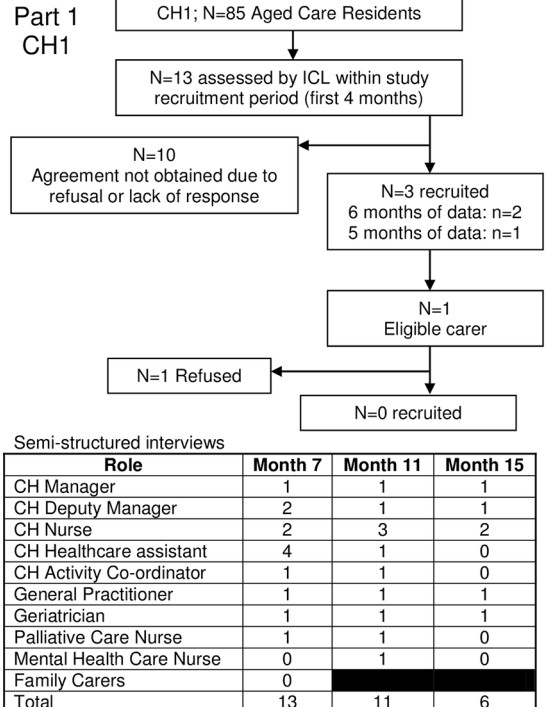

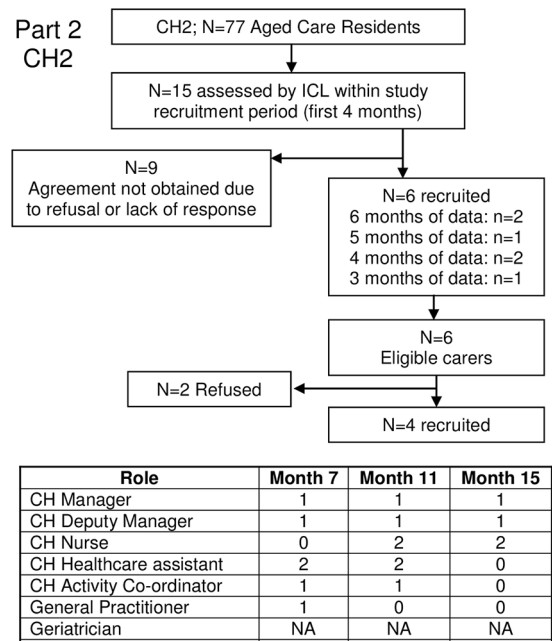

**Semi-structured interviews** (Part 1 CH1)

| Role | Month 7 | Month 11 | Month 15 |
|---|---|---|---|
| CH Manager | 1 | 1 | 1 |
| CH Deputy Manager | 2 | 1 | 1 |
| CH Nurse | 2 | 3 | 2 |
| CH Healthcare assistant | 4 | 1 | 0 |
| CH Activity Co-ordinator | 1 | 1 | 0 |
| General Practitioner | 1 | 1 | 1 |
| Geriatrician | 1 | 1 | 1 |
| Palliative Care Nurse | 1 | 1 | 0 |
| Mental Health Care Nurse | 0 | 1 | 0 |
| Family Carers | 0 | | |
| Total | 13 | 11 | 6 |

(Part 2 CH2)

| Role | Month 7 | Month 11 | Month 15 |
|---|---|---|---|
| CH Manager | 1 | 1 | 1 |
| CH Deputy Manager | 1 | 1 | 1 |
| CH Nurse | 0 | 2 | 2 |
| CH Healthcare assistant | 2 | 2 | 0 |
| CH Activity Co-ordinator | 1 | 1 | 0 |
| General Practitioner | 1 | 0 | 0 |
| Geriatrician | NA | NA | NA |
| Palliative Care Nurse | 0 | 1 | 0 |
| Mental Health Care Nurse | NA | NA | NA |
| Family Carers | 4 | | |
| Total | 10 | 8 | 4 |

**Figure 1** Flowchart of participants.

knowledge and I… No disrespect to the nurses here, I just don't think many of them would have the depth of knowledge and skills to actually do that' (NH1 geriatrician, month 11)

Staff worried about the pressures of time and the need to complete tasks which sometimes meant basic care tasks were overlooked, lengthy discussions about EOL care were impossible and social engagement with residents was minimal.

'Even the patient care, she (ICL) was able to get in and say this one their nails need to be cut, this one has been refusing to get out of bed but their hair needs to be washed, maybe we have applied some approaches but they did not work… (ICL) had all the time, she was able to … give recommendations so actually GP will do this and us (nurses), we'll do this.' (NH1 deputy manager, month 7).

### Activities undertaken

Assessments (Activity 1a), core meetings (Activity 1b) and training (Activity 2) were undertaken in both NHs (table 2). Weekly core meetings were scheduled, but many were cancelled due to staff leave or immediate resident care needs. At NH2, the GP experienced significant time constraints and attended only the first two meetings. The group agreed to weekly meetings with the ICL, manager and nurse with specific medical issues referred to the GP. Core meetings provided an opportunity to discuss individual assessments. These involved the ICL reviewing the resident's file, observing and talking to them and their family, and seeking clarification from NH staff. NH staff had limited time and may have viewed this as duplicating existing assessments. Discussions with

families sought views about current care and concerns about EOL care. The ICL intended to involve NH staff in these discussions but competing staff demands usually prevented this. Common issues identified included swallowing and eating difficulties, pain, pressure area care and lack of social engagement. Advance care plan documentation was more routinely discussed in core meetings at NH1 than NH2.

During core meetings (Activity 1b), staff training needs were discussed and sessions planned (Activity 2), including managing distress during hoist transfers (NH1), and understanding pain and behavioural symptoms (both NHs). At NH1 the manager requested a general information session on dementia and EOL care, while at NH2 the manager requested a half-day session for nurses on pain management and discussing EOL care with family. Fewer training sessions were held at NH2 and staff attendance was suboptimal. Training was positively evaluated (table 3).

Both managers requested the ICL to run information sessions for family members on issues regarding dementia, EOL symptoms and advance care planning. Twelve family members attended at NH1 with the NH manager. At NH2 the session (six families) generated much discussion, overran the allotted time and led to a follow-up session (three families). Evaluations indicated that the sessions were relevant, helpful, contained new information and that the timing was appropriate.

The lower involvement with external healthcare professionals at NH2 prevented establishing wider meetings (Activity 1c). At NH1, 6 months prior to implementation, wider monthly meetings had been initiated. These

**Table 2** Process measures

| Component | Over a 6-month period | NH1 | NH2 |
|---|---|---|---|
| Scoping | ICL visits to NH prior to implementation | 8 | 2 |
| Scoping | ICL visits to external HCPs prior to implementation | 2—palliative care nurse and GP | 0 |
| All components | ICL visits to NH during implementation | 64 | 53 |
| All components | ICL visits to external HCPs during implementation | 1—palliative care nurse | 1—palliative care Lead Clinical Nurse Specialist |
| 1a) Individual holistic resident assessments | Individual assessments completed | 15 | 15 |
| 1a) Individual holistic resident assessments | Number of discussions with family members (not number of family members) | 15 | 24 |
| 1b) Weekly core meetings | Number of meetings | 10 core meetings with GP, deputy manager and nurse from relevant floor (GP missed one meeting) | 8 core meetings with manager and a nurse. GP attended first two meetings |
| 1b) Weekly core meetings | Individualised assessments discussed at core meeting | 15 | 13 |
| 1b) Weekly core meetings | Individual reviews completed | 15 | * |
| 1b) Weekly core meetings | Referrals made to external HCPs | 6 (2 × community mental health team; 2 × speech and language therapist; 2 × occupational therapist) | 4 (3 × old age psychiatrist; 1 × manual handling trainer) |
| 1c) Monthly wider team meetings | Number of meetings | 6 meetings; usually with geriatrician, GP, palliative care nurse, Triage and Rapidly Elderly Assessment Team, NH nursing staff and deputy manager (and/or manager) | Wider meetings not established. The ICL was able to arrange one meeting with the palliative care nurse, NH manager and deputy manager |
| 1c) Monthly wider team meetings | Number of residents assessed by ICL discussed | 11 | Not applicable |
| 2) Education | Number of training sessions (total number of attendees) | 9 (84) | 5 (21) |

*No formal reviews involving reassessment were completed at NH2, although there was subsequent discussion of many of the residents at subsequent meetings.
GP, general practitioner; HCP, health care professional; ICL, interdisciplinary care leader; NH, nursing home; NH1, nursing home 1; NH2, nursing home 2.

meetings were supported by the ICL and involved both review of residents requiring palliative care and reflecting on whether EOL care processes could have been better for deceased residents.

### Implementation phases

The staff and family interviews give information on the five implementation phases.[23]

#### Phase 1: Orientation

NH managers highlighted their role in promoting the Intervention; '*Within two or 3 weeks I had gone in and prepared the staff that she (ICL) was going to be here and that she had full access to the records and the staff*' (NH1 manager, month 7). Staff and family engagement was attributed to the importance of the Intervention topic. '*I am happy that something like this is going on, that someone is interested and is trying to help people with dementia and end of life*' (NH1 nurse, month 7); and '*I think it was right for the programme to suggest and talk about end of life palliative care*' (NH2 family carer, month 7). Characteristics of the ICL were attributed to engaging staff with the Intervention; '*(ICL) was very helpful… I would say she's a very good listener… she's got plenty of time, which I think is lovely*' (NH2 deputy manager, month 7).

#### Phase 2: Insight

As reported under context, NH staff had only basic knowledge regarding dementia EOL care and it was important that they gained insight into the need for practice improvements. Many staff were receptive to receiving information. Training from the ICL improved knowledge and promoted a person-centred view of dementia care. The Intervention provided insights into existing routines critical for driving practice improvements, often highlighting existing deficits in the care being provided:

**Table 3** Staff training evaluation

| NH | Reducing distress during personal care | Behaviour and pain management | | EOL care in dementia | |
|---|---|---|---|---|---|
| | NH1 (n=23) | NH1 (n=36) | NH2 (n=12) | NH1 (n=25) | NH2 (n=9*) |
| Duration in hours | 1 | 1 | 1 | 1 | 4 |
| Sessions | 2×day; 1×night | 2×day; 1×night | 2×day; 1× night and day | 2×day; 1×night | 2×nursing staff |
| **Evaluation: median (IQR)** | | | | | |
| Was this training relevant to your day-to-day work?† | 4 (3–4) | 4 (4–4) | 4 (3–4) | 4 (3–4) | 4 (3.25–4) |
| Did you learn anything new from the training?† | 3 (3–4) | 4 (3.25–4) | 4 (3–4) | 3 (3–4) | 3.5 (3–4) |
| Do you think this training will influence your work?† | 4 (3–4) | 4 (4–4) | 4 (3–4) | 3 (3–4) | 4 (3–4) |
| What was the training level?‡ | 1 (0–1) | 1 (1–1) | 1 (1–1) | 1 (1–1) | 1 (1–1) |
| Did the training provide a useful refresher?† | 3 (3–3) | 4 (3–4) | 3 (3–3.75) | Not asked | Not asked |
| Has this training improved your confidence in talking to family about EOL care?§ | Not asked | Not asked | Not asked | 4 (4–4) | 4 (4–4) |

*Evaluation sheet missing from one attendee.
†Measured on a 5point Likert Scale from 0=strongly disagree to 4=strongly agree.
‡Measured on a 3-point Likert Scale: 0=too basic; 1=about right; 2= too complex.
§Measured on a 5point Likert Scale from 0=not at all to 4=yes, a lot; higher median better.
EOL, end of life; NH, nursing home; NH1, nursing home 1; NH2, nursing home 2.

'… through these 6 months I realised… the paperwork was being reviewed, reviewed, reviewed but actually the patient was not being reviewed it was just being carried forward.' (NH1 GP, month 7).

'She needs to give us more training about the caring, like dementia. It will also help us communicate with our colleagues because some of our colleagues don't know how to communicate with the service user; she can train them how to do it.' (NH1 healthcare assistant, month 11).

'I think we will take on her advice that she gave on end of life and on dealing with dementia for the relatives. We deal with the residents but then it's the relatives that… need the help. Why's this happening? Why doesn't he know them? We do a lot with the residents but not with the relatives.' (NH2 activity coordinator, month 7).

While wider meetings at NH1 had started before implementation, the ICL also provided an alternative view during these meetings:

'…her (ICL) input was useful… during the MDM (wider multidisciplinary meeting)…her feedback and some of her suggestions actually helped us to see things a little bit differently' (NH1 geriatrician, month 7).

### Phase 3: Acceptance
Staff were energised by the Intervention as it provided an opportunity to develop new ideas and skills, and, ultimately, improve dementia care:

'… anybody new coming (in) will come up with new ideas, new experiences from other places, it's building

up. You cannot say I am that clever when I am not. I am open to new ideas all the time.' (NH1 nurse, month 7).

'I never knew what it was she (ICL) was willing or she was about to tell me. But because it was end of life management I hope it is good for every carer to know how to manage… it will help me to get some ideas to prepare and how to deal with those situations.' (NH1 healthcare assistant, month 7).

However, initially, the NH staff were wary of change and the ICL experienced some early difficulties engaging:

'I don't know that the staff really understood for quite a while why she (ICL) was there and what she was doing. I don't think it was her problem; I think it was more what the project was all about.' (NH1 palliative care nurse, month 7).

### Phase 4: Change
Participants identified practices that had become part of NH protocols and routines as a result of the Intervention. Participants confirmed the value of the ICL's EOL discussions with family carers. At NH1 a modified template to support advance care planning was introduced to replace three existing care plans relating to EOL wishes, and to provide greater guidance to NH staff about how to manage possible EOL symptoms. At NH2 modifiable wall-mounted care charts (Care Charts UK ©) in residents' rooms were introduced to communicate residents' needs and preferences. Greater focus on pain assessment for residents who were unable to verbally

communicate led to introducing the Pain Assessment in Advanced Dementia Scale [34] and pain management plans at NH2.

'(ICL) gave me this wonderful sheet about pain control, really and how to… so we've implemented some of the things that she has given to us.' (NH2 deputy manager, month 7).

However, time demands also prevented NH staff and GPs attending Intervention meetings and training:

'It was really good what she was saying but I haven't got the time to do it. So she would sit and discuss them and it would take them half an hour forty minutes to talk about two or three patients and if I've got to see fourteen in the morning - I just can't do it.' (NH2 GP, month 7).

'I didn't do the end of life training; not that I didn't want to do it, there was not really the chance to go in there.' (NH1 healthcare assistant, month 7).

### Phase 5: Maintenance

Staff described the new Advance Care Plan at NH1 and pain management plans and the wall mounted care charts at NH2 as being maintained at months 11 and 15 and becoming embedded into routine care:

'The care (nursing) home are actually using her template, developed a new advanced care plan which has incorporated the points that she (the ICL) raised and so that's what we are using now, for all new patients that come in… existing patients, we are transferring gradually'. (NH1 GP, month 11).

'Do you know who loves them (care charts) best? Can I tell you, the relatives… they will tell you the detail about their loved one… So the minute somebody comes in I tell them about the work that the ICL did and then I tell them about the 'this is me' life profile… when we had our Care Quality Commission inspection they really liked the 'this is me' profiles' (NH2 manager, month 15).

It was apparent that the need for staff development and a shift from task-driven to compassionate care would require a longer duration and further training and support from the ICL. Continuing support and training from the ICL could build on this work, further enhancing staff confidence.

'I think that if she'd been there for a whole lot longer then what would have happened is there would be an evolving of her role in a sense that the issues that were raised would have become identified by the nurses as routine.' (NH1 GP, month 7).

### Cost of implementation

presents the time the ICL spent on various activities and this was used to calculate the costs of Implementation. Of the total 656 hours, 42% were spent on NH1 activities, 34% on NH2 activities and 24% on activities not attributable to one particular NH. Engagement of the ICL to implement the Intervention in two NHs for 6 months was costed at £18 255 including on-costs and travel fares (and excluding time the ICL spent on non-Intervention activities).

### Individual resident and carer outcomes

We recruited 9/28 residents assessed by the ICL for monthly data collection (figure 1). Recruitment was hampered by difficulties engaging with family members who had limited day-to-day involvement with their relative and did not respond to letters and phone calls. Four residents died or moved NH before agreement was obtained. One daughter declined participation due to her family's request that their relative should not be involved in research.

At NH1 the three residents had a median age of 81 years (IQR: 76–93) and two were female. At NH2 the median age of the six residents was 80 years (IQR: 76–85) and all were female. Data were descriptively compared with those from the larger cohort (table 4). As none of the 9 participants died during the data collection period, we compared their outcomes with the 52 participants involved in the cohort study who survived the 9-month data collection period. Findings in table 4 suggest that the Intervention did not cause harm to residents, but the effects on carers at NH2 may need further consideration.

### NH-wide outcomes

NHs did not maintain electronic records of any of the NH-wide outcomes. Manual searches of daily logs and individual care plans were required. At NH1 resuscitation status was not documented consistently and at NH2 obtaining these data required reading of individual care plans. Due to these difficulties we reduced collection frequency to three time points (months 1, 4 and 7). What data were collected showed few of out-of-hours GP calls and visits, ambulance calls and unplanned hospitalisations. At NH1 pain management plan frequency increased slightly during implementation from 71% to 85% of residents. Preferred place of death was reported for 30% of residents at month 1 and 85% at month 4 (month 7 data were unavailable). These measures could only be collected at month 1 in NH2 where we found one resident (not cognitively impaired) had a pain management plan in place, 21% had their preferred place of death recorded and 30% had a documented 'Do not attempt resuscitation' form.

Over the 7-month data collection period, 17 NH1 residents died, 10 in their usual NH. For the seven hospital deaths, one was the preferred place of death reported by family and another did not have a documented preference. For two residents with the NH documented as the preferred place, families requested their relative be admitted to hospital. At NH2 for the 3 months in which resident deaths were reported, 12 residents died and 7 who had a documented preference, died in their preferred place.

## DISCUSSION
### Principal findings

We report on how the Compassion Intervention operated in two UK NHs in different healthcare funding

**Table 4** Resident and carer evaluation data compared with a larger cohort

| Baseline assessment | Cohort study (n=52)* | NH1 (n=3) | NH2 (n=6) |
|---|---|---|---|
| **Functional Assessment Staging Scale** | | | |
| 6b–6d (Unable to bathe independently— urinary incontinence) | 0 | 0 | 1 |
| 6e–7b (doubly incontinent—loss of ability to speak >6 words) | 21 | 1 | 4 |
| 7c–7e (ambulatory ability lost—can't hold up head independently) | 31 | 2 | 1 |
| **Charlson Comorbidity Index** median (IQR) | 6 (6–7) | 6 (4–7) | 5 (4–6) |
| **Bedford Alzheimer Nursing Severity Scale** median (IQR) | 22 (18–23) | 22 (21–24) | 22 (20–23) |
| *Final visit* | *Cohort study (n=52)* | *NH1 (n=3)* | *NH2 (n=6)* |
| **Waterlow Scale (pressure ulcer risk)** | | | |
| High risk (15-19) | 14 (27) | 1 (33) | 1 (17) |
| Very high risk (≥20) | 36 (69) | 2 (67) | 4 (67) |
| **Neuropsychiatric inventory**—number of symptoms, median (IQR) | 4 (1.5–6) | 2 (2–5) | 4 (2–6) |
| **Cohen-Mansfield Agitation Inventory:** behavioural disturbances (≥39) | 29 (56) | 1 (33) | 3 (50) |
| **Pain Assessment in Advanced Dementia Scale:** (n, %) | | | |
| Rest (≥2) | 10 (19) | 0 (0) | 2 (33) |
| Movement (≥2) | 29 (60) | 2 (67) | 1 (17) |
| **Symptom Management at EOL in Dementia Scale** median (IQR) | 26 (20–35) | 30 (26–32) | 33 (31–37) |
| **Quality of Life in Late Stage Dementia Scale** median (IQR) | 24.5 (20–28.5) | 23 (23–31) | 25 (20–28) |
| *Carer measures:* | (n=23) | (n=0) | (n=4) |
| **Zarit Burden Interview** median (IQR) | 11 (6–18) | | 23 (15–28) |
| **Hospital Anxiety and Depression Scale ≥8** n (%) | | | |
| Anxiety | 8 (35) | | 2 (50) |
| Depression | 5 (21) | | 2 (50) |
| **Satisfaction with Care at EOL in Dementia Scale** median (IQR) | 30 (29–33) | | 34 (28–39) |
| **Resource Utilization in Dementia Questionnaire** median (IQR) | | | |
| Visits from doctor, physiotherapist, psychologist, other HCP in previous month | 1 (1–3) | 0 (0–2) | 1 (1–2) |
| All general hospital admissions in previous month | 0.5 (0–1) | 0 (0–0) | 0 (0–0) |

Charlson Comorbidity Index (19 diseases)[29]
Bedford Alzheimer Nursing Severity Scale: range 7–28, higher scores indicate severity[30]
Waterlow Scale: range 2–46, higher score higher pressure ulcer risk[31]
Neuropsychiatric Inventory: total symptoms, maximum 12[32]
Cohen-Mansfield Agitation Inventory: range 29–203, scores ≥39 indicates clinically significant agitation[33]
Pain Assessment in Advanced Dementia Scale: range 0–10; scores ≥2 indicates pain[34]
Symptom Management at EOL in Dementia: range 0–45; higher scores indicate better symptom control[35]
Quality of Life in Late Stage Dementia Scale: range 11–55, lower scores indicate better quality of life[36]
Zarit Burden Interview: range 0–88, higher scores indicate greater burden[37]
Hospital Anxiety and Depression Scale: Anxiety and depression subscales range 0–21, scores ≥8 indicate clinically significant depression or anxiety[38]
Satisfaction with Care at EOL in Dementia: range 10–40; higher scores indicate more satisfaction with EOL care[35]
Resource Utilization in Dementia Questionnaire[39]
*The cohort study involved 85 residents in total but this table only includes the 52 participants who survived the 9-month data collection period.
EOL, end of life; HCP, health care professional

systems and the feasibility of implementation. Our data inform evaluative studies to address gaps in EOL care for residents with advanced dementia. We found that implementation was dependent on several aspects of the local NH context. These included the state of readiness for accepting the intervention, in particular local funding priorities within the healthcare system and relations between multidisciplinary care providers across specialist and generalist services; organisational structures within the NH including staffing levels, confidence, knowledge and skills of staff, and existing assessment procedures for residents. The period of implementation was short but there was evidence that the Intervention achieved acceptance within both NHs. We noted changes in care

processes such as advance care planning, pain management and the introduction of wall-mounted care charts; these were maintained 9 months later. Despite limited NH staff availability, three of the four key activities were implemented in both NHs. No wider meetings and fewer training sessions were implemented at NH2 than NH1. The NH context may explain these differences.

We were unable to assess whether changes led to better outcomes for residents or family, but there were no indications of harm to residents. Of concern was that the small number of carers recruited appeared to have poorer mental health when compared with the wider cohort, despite reporting benefits of participation and higher satisfaction with end-of-life care. Possibly distressed carers seeking support were more motivated to participate. Previous studies suggest that EOL discussions can improve carer satisfaction with EOL care.[44] We have analysed practice relating to EOL conversations elsewhere.[28]

### Strengths and weaknesses

This was an exploratory study. While the sample size was small, we did not aim to detect differences or calculate a sample size for future studies. Our work is strengthened by the theory and evidence underpinning the Intervention described in earlier publications.[6 21 27] We took note of contextual factors affecting the five phases of implementation described in the literature on whole systems change in healthcare.[23] Our Intervention provides a framework that may promote EOL care in accordance with EAPC recommendations.[5] The Compassion Intervention supports many of the EAPC's domains including: (2) person-centred care, communication and shared decision making; (3) setting care goals and advance planning; (6) avoiding overly aggressive, burdensome or futile treatment; (7) optimal treatment of symptoms and providing comfort; (8) psychosocial and spiritual support; (9) family care and involvement; and (10) education of the healthcare team.

Our implementation phase was short. There was limited time for the ICL to gain the trust of key stakeholders and family members. The short time frame and the difficulty in scheduling weekly meetings to discuss assessments limited the number of residents who could be assessed and who were therefore eligible for recruitment for collecting individual outcome data. Often the person listed as a proxy decision maker had minimal contact with the resident and felt unable to make decisions on their behalf, prohibiting recruitment of both carers and residents. Using professional consultees enabled involvement of isolated residents.

Recruitment of only four informal carers limits our understanding of the impact of the Intervention on families and this needs exploration in future work. There is evidence from other research[44] that carers do benefit from attempts to improve care for relatives with dementia who are dying.

Involvement of the ICL in both roll-out and monitoring of the Intervention (KJM) creates potential for bias. This may be counterbalanced by the depth of understanding achieved which was of importance at this stage of evaluation. We engaged independent researchers in the analysis of interviews (NK, GL) and quantitative data (AG, VV, RZO, ELS) and all coauthors critically reviewed the findings. We have not incorporated an analysis of the ICL diary here, but autoethnographical findings have been published elsewhere.[28]

### Implications and future research

Consistent with previous studies,[45] collecting NH-level data proved challenging and further evaluations should allocate resources for collecting reliable data. The low frequency of deaths, unplanned hospitalisations and out-of-hours calls implies a large number of NHs would be required to give sufficient power to investigate NH-wide outcomes. These measures are not very sophisticated indicators of quality of end-of-life care and individual resident measures may be more useful as they describe symptom burden. The Symptom Management at EOL in Dementia[35] and the Satisfaction with Care at EOL in Dementia[35] Scales can assess multiple EOL symptoms and family satisfaction with care.

The criteria for inclusion may appear inappropriate given that none of the recruited residents died during the intervention period. However, three had died in the period between the ICL assessment and the research team trying to recruit the participant. In addition, another participant died a few weeks after the Intervention period ceased. The other deaths in the NHs were among residents who did not all have dementia. Also, there were residents who were eligible for the Intervention but who the ICL had not had time to assess during the Intervention period. Also, our larger cohort study,[21] using similar eligibility criteria found that only 36% of residents with advanced dementia died during a 9 month observation period, reflecting the difficulty in prognosis of EOL in dementia. We advocate a proactive approach to addressing advance care planning and actively managing symptoms of pain and discomfort for all NH residents, with the need for particular attention to the unique needs of residents with advanced dementia and limited capacity to verbally communicate their needs.

We have information regarding the costs, time and skills required to engage an ICL. We also highlight the benefits of an ICL who was external to the NH to drive practice change and to provide independent support for family carers.[46] For localities with good external multidisciplinary support for NHs, the Intervention might be implemented by employing a full-time ICL working across two to three NHs. However, for contexts such as NH2, external support from a range of disciplinary areas (not costed in this study) would require greater investment from commissioners. The extent to which the context of NH1 or NH2 reflects the typical level of support for UK NHs is unknown.

Further investigation of the Intervention could examine how the ICL role might be integrated into

usual practice, perhaps upskilling an existing NH staff member, harnessing the expertise of a member of the wider multidisciplinary team or through palliative care services provided within the charitable sector such as outreach from a hospice. The benefits of external facilitation from programmes such as the Gold Standards Framework have been demonstrated for supporting end-of-life care in NHs.[47] The ICL may be challenged by working across a large number of NHs and flexibility is needed to allow enough time within each NH for the ICL to integrate and be effective. Further work is required to determine whether the ICL role would need to remain at the same level of intensity and for how long. There is the need for someone with the skills to discuss end of life with family carers and to provide staff training, given the high turnover of direct care staff in NHs.[48] During family group sessions it was evident that carers had a poor understanding of dementia and wanted to learn about all aspects of dementia, not only about EOL. Staff in the facility lacked confidence in providing information to families and would require a considerable amount of development in EOL dementia care before a role of an ICL became redundant.

Our ICL had a social care background but individuals with a different disciplinary background, such as a palliative care nurse or dementia-specific Admiral Nurse, may have brought different skills to the role and focused on different goals and care issues. A key benefit of Compassion appeared to be the ICL offering a more holistic approach which went beyond physical and medical care needs. Professional development and clinical support for the ICL role was crucial.

Further work also needs to examine the applicability of the model to long-term care settings where nursing care is not available. We focused on NHs in this study as residents fitting the criteria for advanced dementia would most likely require NH-level of care. In this study we did not involve healthcare assistants in core or wider meetings although their input was sought during assessments and they received training to improve EOL knowledge.[49] The benefit of involving them in the core and wider meetings requires further investigation.

Our work did not lead to substantial changes to the Compassion Intervention manual. The assessment template we developed aimed to be holistic covering a broad range of issues including the person's physical, social, psychological and spiritual needs. Although observational assessments may have identified environmental factors that impacted on the resident's well-being, these were not explicitly included in the assessment but could be important to include.[50] Further testing of the Intervention may lead to further refinement of the assessment and identify new elements over time. In addition, the assessment required some duplication of existing assessments undertaken in each NH. To address this issue we have added a checklist to prompt NHs to examine existing assessment domains rather than requiring another assessment template. Prior to working with this Intervention,

NHs should consider the feasibility of weekly core meetings and how to incorporate assessments into existing processes.

The Compassion Intervention was underpinned by organisational change theory.[23] There have been few EOL intervention studies developed in NHs in advanced dementia, but none that have used an external role such as an ICL to facilitate practice change. External facilitators of the education-focused GSFCH reported concerns about their lack of time to enable adequate support.[51] The level of facilitation in the Compassion Intervention was higher than the 'high facilitation' reported in the GSFCH programme, and training on its own is unlikely to change resistant norms and practices.[20] The study using the most similar approach but has not been completed at the date of this paper may provide useful insights into the benefits of family case conferencing in the NH setting[17] with implementation of a similar role as ICL but from a nurse within the NH. This will provide a useful comparison for the importance of an internal or external ICL.

Our implementation was flexible in responding to the unique needs of the different NH contexts and the holistic assessments undertaken by the ICL were crucial in providing insights to NH staff about gaps in existing care provision. The ICL implemented a relationship-centred approach which aimed to provide information and practical and emotional support to NH staff, family and residents.[52] However, other approaches to implementing practice change are also worth considering. For example, action research used in the NH setting has been useful in transforming task-driven approaches to approaches that engage staff more meaningfully with care processes to enable practice improvements.[53]

## Conclusion

Implementation of the Compassion Intervention was feasible to differing degrees across two sites, dependent on context. The role of the ICL appeared the key factor for supporting practice change in this exploratory study. Our data inform future testing to identify the Intervention's effectiveness in improving end-of-life care in advanced dementia.

**Twitter** @MCPCRD

**Acknowledgements** The authors thank other members of the Compassion research team, past and present, for their support in completing this research, particularly Sharon Scott and Steve Morris for their earlier contributions in developing the Compassion programme, and Margaret Elliott for her work on developing the Compassion Intervention manual, providing clinical support to the ICL, and undertaking qualitative interviews. The authors also thank Ritchard Ledgerd and Dr Karen Harrison-Dening for providing clinical support to the ICL outside the research team, Professor Martin Marshall for advice on implementation theory, and Marie Curie for funding and supporting the programme, the two care homes that completed the Intervention and the health care professionals (HCP), residents and carers who participated in this study.

**Contributors** All authors made substantial contribution to the conception or design of the work or the acquisition, analysis, or interpretation of data for the work. All authors were involved in drafting the work or critically revising it, approving the final version to be published and agree to be accountable for the work.

**Funding** This work was supported by Marie Curie Cancer Care (now Marie Curie), (grant number: MCCC-FPR-11-U) through a process administered in partnership with Cancer Research UK.

**Competing interests** Authors had no competing interests

**Patient consent** Detail has been removed from this case description/these case descriptions to ensure anonymity. The editors and reviewers have seen the detailed information available and are satisfied that the information backs up the case the authors are making.

**Ethics approval** Ethical approval for roll-out of Compassion and data collection was granted by the National Research Ethics Service, London—Camden and Islington Research Ethics Committee (Reference 14/LO/0370) and for assessment of maintenance and sustainability by UCL Research Ethics Committee (ID 3618/001).

**Provenance and peer review** Not commissioned; externally peer reviewed.

**Data sharing statement** All process data is included in the paper. We do not have ethical permission to disseminate interview transcripts.

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
