## [Reviewer comments · BMJ Open]

ARTICLE DETAILS

TITLE (PROVISIONAL)	Implementing the Compassion Intervention, a model for integrated care for people with advanced dementia towards the end of life in nursing homes: A naturalistic feasibility study
AUTHORS	Moore, Kirsten; Candy, Bridget; Davis, Sarah; Gola, Anna; Harrington, Jane; Kupeli, Nuriye; Vickerstaff, Victoria; King, Michael; Leavey, Gerard; Nazareth, Irwin; Omar, Rumana; Jones, Louise; Sampson, Elizabeth

VERSION 1 - REVIEW

REVIEWER	Julie Watson Edinburgh Centre for Research on the Experience of Dementia School of Health in Social Sciences University of Edinburgh UK
REVIEW RETURNED	03-Mar-2017

GENERAL COMMENTS	Overall This interesting paper tackles one of the most important issues of our time, the end of life care of people with advanced dementia in care homes. It describes the processes and outcomes of a feasibility study of the 'Compassion Intervention' facilitated by an Interdisciplinary Care Leader in two care homes. The findings are important, not only for the Compassion Intervention' and how it proceeds based on the findings, but also for others who are grappling with this issue and trying to find solutions. Specific Comments on Sections Introduction It would be helpful to put the Compassion Intervention in the context of other work that has been done in this area that is referred to on Page 26 Lines 21-42 and draw out what is different or unique about the Compassion Intervention and also what is similar or overlapping with other work. I'd like a clearer sense of what the intervention would entail e.g. I'm not clear if this intervention was ultimately commissioned would the ICL be there permanently or just for a period of time? If a central care co-ordinator is seen as a key part of taking an integrated approach as you suggest on page 4, more explanation needs to be given as to why in your feasibility study the ICL was there for only 6 months and you then measured the sustainability of the changes made. Also a clearer outline of the competencies required by an ICL would be helpful, perhaps with reference to this paper: Stanyon, M.R. Goldberg, S.E. Astle, A. Griffiths, A. Gordon, A.L. (2017) The competencies of Registered Nurses working in care homes: a modified Delphi study Age and Ageing doi: 10.1093/ageing/afw244 Table 1
---

1a

I think it would be helpful to clarify what assessment of EOL wishes entails – in other parts of the paper you talk about resuscitation status, preferred place of death and I think they should have a clearer presence here.

It strikes me that the ‘voice’ of the person with dementia is not reflected in this assessment template – I would like to know how what they are communicating, either verbally or in embodied ways, such as spitting food out, resisting care, or hugging and smiling, for example, is taken into account.

Also families of people with advanced dementia often have a high level of support needs and that is not clearly reflected in this assessment template.

Page 5 – it might be helpful to include a box on the components of compassion you mention to give the readers a better understanding of what you mean by compassion.

Page 7 – ethics might be better in here after aim – it would flow better.

Page 8 - line 46/47 – to clarify – is this component 1a of the intervention? A clearer link in the text to Table 1 would help readers.

Page 11 – line 34 – should this be Table 2?

Page - 13 line 32 – what was the reason for not including residents from the residential unit given that there now seems to be very little difference between the needs of residents in care homes with and without nurses e.g. Bowman and Meyer Age and Ageing 2017; 0: 1–2

doi: 10.1093/ageing/afx030

Page 15 – it would be helpful if there was more clear linkage in this table to the key activities in Table 1 e.g. Core meetings – is that 1b? Did each care home have a specified GP from one practice, or a lot of practices caring for different residents? Please clarify as this has implications for the commissioning of the Compassion Intervention in the future.

Page 15 - In Table 1 it says that those delivering the social care (care assistants?) were involved in component 1b but they are not mentioned in Table 2 as being present at core meetings– some discussion of why this was and the implications of this would be useful in the discussion section.

Table 2 – Individual reviews completed – what does this relate to in Table 1?

Page 17 – information sessions for families – if sustainability was important would it have been better to train up the staff in this? - needs some reflection and discussion.

Page 17 - Line 14 – what was the generated discussion about? This would be useful for readers to know as it would give some insight into families’ experiences – unless it is confidential

Page 19 - Line 42 – unable to verbally communicate – still communicating in embodied ways

In the qualitative data reported there is an absence of any quotes from healthcare assistants – I would like to hear from them too to get a more in-depth understanding of the processes and outcomes from their perspective and in terms of the rigor of the analysis.

Page 23 – the reason why NH wide outcomes were not easily obtained needs some explanation.

Page 23 – Line 46-57 – Are these indicators of success? If so it would be helpful to mention them earlier in the paper.

Page 24 – were some of those who died people who were eligible to take part in the study but whose families had not given consent?

Were there other people who died who had not met the eligibility criteria? I think it would be useful to reflect on some of this in the

discussion as it is relevant for the next stage of the Compassion Intervention.
Page 25 Line 7/8 – carers mental health – I think this is the first mention of this - I think this needs some clarification of where and why it was measured and the potential implications of it.
Page 25 - Line 23/24 – I think you need to present a much stronger argument based on the data you have collected to justify why you think your intervention provides a framework to optimise EOL care in accordance with the EAPC recommendations.

Discussion

I would like to see a much more explicit account of what the authors have learnt from doing this feasibility, what worked well and more detail on how the findings will inform future testing. What comes across is that it was a struggle to implement in both care homes. This is important work to report but the implications of it need more discussion. There needs to be more engagement with the challenges e.g. low recruitment of residents from among those eligible – what might have helped with this? What can be learned about recruitment from other care home studies or organisations such as ENRICH? Are weekly core meetings realistic? More information about the cost-effectiveness of the intervention would be helpful. Given that there were deaths in both care homes but not among those in the study, were the eligibility criteria for recruitment right? The criteria on page 5 for inclusion could describe many care homes residents who might live with that level of need for many months. What are the implications of this for how we define end of life care for people with dementia and what it entails and the staff competencies and staffing levels required? The authors say on page 25 Line 23-27 that this intervention provides a framework to optimise EOL care in accordance with EAPC recommendations but I would like them to present a more convincing discussion on why they believe this to be the case based on the results presented.

Page 20 lines 52/53 touches on the need to change the culture from a task driven culture to a compassionate care culture. It would be helpful if the authors could discuss in light of the feasibility study whether they think their approach is the right approach to changing cultures of care with reference to other literature on developing practice and changing cultures of care e.g.

Froggatt, K. Davies, S. & Meyer, J. (eds) (2009) *Understanding Care Homes: A Research and Development Perspective* Jessica Kingsley Publishers: London

McCormack, B and McCance, T. (2017) *Person-centred practice in nursing and healthcare* Wiley Blackwell.

I'd like to see some discussion of whether the authors consider the assessment template to be comprehensive enough. Environmental and design issues such as excess noise and bright lights can be confusing for people with dementia and lead to agitation and poor quality of life. Were these factors considered as part of the intervention, or considered beyond its scope? If so it would be helpful to discuss them as a limitation with reference to the wider literature e.g.

Chaudhury, H. Puurveen, G. & Lyle, J. (2011) *Place matters: an exploration of the role of physical environment in end of life care* Chapter 17 in Gott, M. & Ingleton, C. (eds.) (2011). *Living with Ageing and Dying: Palliative and End of Life Care for Older People* Oxford University Press: Oxford

Chaudhury, H. Hung, L. & Badger, M (2013) *The Role of Physical*

	Environment in Supporting Person-centered Dining in Long-Term Care: A Review of the Literature American Journal of Alzheimer's Disease and other Dementias 28(5) 491-500 Clarke, Work has been done in care homes that shows that systems on their own are not enough to change practice but attention also needs to be paid to the lifeworld of the staff, not just in relation to training but also in terms of emotional support to deal with the nature of caring for very frail people at the end of life e.g. Froggatt, K. Hockley, J. Parker, D. & Brazil, K. (2011) A system lifeworld perspective on dying in long term care settings for older people: Contested states in contested places Health and Place 17 263-268. Some reference to this wider work would strengthen the discussion greatly.
--	--

REVIEWER	Sandra Oppikofer University of Zurich Center for Gerontology Dynage Switzerland
REVIEW RETURNED	08-Mar-2017

GENERAL COMMENTS	This manuscript reports an interesting and highly relevant study with practical implications for EOL-patients, even though the n is small. The paper is well written and the study has several strengths and will be an important contribution to the field of EOL and dementia.
--

VERSION 1 – AUTHOR RESPONSE

We thank the two reviewers for considering this paper and for seeing the importance of developing interventions to improve end of life care in dementia. The comments from reviewer 1 enabled us to reincorporate some of the discussion we had previously discarded due to concerns about paper readability and length. Please find our response to each of their comments below.

Reviewer comment: It would be helpful to put the Compassion Intervention in the context of other work that has been done in this area and draw out what is different or unique about the Compassion Intervention and also what is similar or overlapping with other work.

Response: We have added a paragraph in the introduction describing in more detail existing EOL programmes developed in care home settings and in advanced dementia. Under the description of the Intervention we have also added: "The Intervention has similar components to existing EOL programmes in care homes such as education provision [43 44] and multidisciplinary input (Agar, 2015). The key distinguishing feature of the Intervention is the role of the independent ICL who works solely with two NHs to provide mentoring, role modelling, advice and training and who can develop relationships with NH staff, residents and family carers and develop an in-depth understanding of the organisational culture underpinning practice and impacting on practice change initiatives."

Reviewer comment: I'd like a clearer sense of what the intervention would entail e.g. I'm not clear if this intervention was ultimately commissioned would the ICL be there permanently or just for a period of time? ...more explanation needs to be given as to why... the ICL was there for only 6 months and you then measured the sustainability of the changes made.

Response: In the limitations we have added “Our implementation phase was short. There was limited time for the ICL to gain the trust of key stakeholders and family members.” In the implications section we have added “Further investigation of the Intervention could examine how the ICL role might be integrated into usual practice, perhaps up-skilling an existing NH staff member, harnessing the expertise of a member of the wider multidisciplinary team or through palliative care services provided within the charitable sector such as outreach from a hospice. The benefits of external facilitation from programmes such as the Gold Standards Framework have been demonstrated for supporting end of life care in NHs (Seymour, 2011). The ICL may be challenged by working across a large number of NHs and flexibility is needed to allow enough time within each NH for the ICL to integrate and be effective. Further work is required to determine whether the ICL role would need to remain at the same level of intensity and for how long. There is the need for someone with the skills to discuss end of life with family carers and to provide staff training, given the high turnover of direct care staff in NHs (22). During family group sessions it was evident that carers had a poor understanding of dementia and wanted to learn about all aspects of dementia, not only about EOL. Staff in the facility lacked confidence in providing information to families and would require a considerable amount of development in EOL dementia care before a role of an ICL became redundant.”

Reviewer comment: Also a clearer outline of the competencies required by an ICL would be helpful, perhaps with reference to this paper..

Response: In the introduction section we have expanded the description of the role of the ICL to state: “The ICL role requires a broad range of skills including knowledge of dementia and end-of-life care, ability to educate staff and talk empathically with family carers, and sensitivity to identify and minimise poor care practices.”

Reviewer comment: Table 1: 1a: I think it would be helpful to clarify what assessment of EOL wishes entails... Also families of people with advanced dementia often have a high level of support needs and that is not clearly reflected in this assessment template.

Response: Next to ‘EOL wishes’ we have added: “(Did the resident document preferences when they had capacity? Are family carer preferences documented? Are resuscitation status and preferred place of death documented and reviewed?)” Under ‘who is involved’ it states that “The process involves liaison with the resident and family about their perceived needs, issues and expectations regarding EOL care.” The ICL had an informal discussion with the carer rather than a formal assessment of their needs. The ICL discussed with the carer the current care being provided to the resident, whether they had any concerns about that care, whether they wanted to discuss or amend EOL documentation and how they were coping.

Reviewer comment: Table 1: 1a: It strikes me that the ‘voice’ of the person with dementia is not reflected in this assessment template...

Response: We have added under ‘Who is involved’ in section 1a: “The assessment involves observations and if possible, discussions with the resident. The assessment template focuses on observational measures to identify whether the resident is showing signs of comfort, discomfort, distress and/or pain.”

Reviewer comment: Page 5 – it might be helpful to include a box on the components of compassion you mention to give the readers a better understanding of what you mean by compassion.

Response: The components are described in detail in Table 1. On page 5 we have moved the statement “There are two core components: facilitation of an integrated, multi-disciplinary approach to assessment, treatment and care; and education, training and support for formal and informal carers”

to the start of the next paragraph and have added another reference to Table 1 to make the connection clearer.

Reviewer comment: Page 7 – ethics might be better in here after aim – it would flow better.

Response: We disagree with moving the ethics section to before the method section. We have included ethics where most academic papers include it. Without having read the method section it would be unclear that there were two separate levels with different ethical requirements. The ICL undertaking assessments of eligible residents was part of the intervention and only required consent by the care home whereas evaluation of the intervention required consent at the level of the individual resident and therefore was a separate process. This would be confusing to include at the start of the method section.

Reviewer comment: Page 8 - line 46/47 – is this component 1a of the intervention? A clearer link in the text to Table 1 would help readers.

Response: We have added more links throughout the text to the activity number from Table 1.

Reviewer comment: Page 11 – line 34 – should this be Table 2?

Response: No, it is referring to Table 1. To make it clearer we have reworded the sentence to: “Process data are reported as total number of activities (as outlined in Table 1) undertaken and total ICL hours spent on different activities.”

Reviewer comment: Page - 13 line 32 – what was the reason for not including residents from the residential unit given that there now seems to be very little difference between the needs of residents in care homes with and without nurses

Response: In implications we have added: “Further work also needs to examine the applicability of the model to long term care settings where nursing care is not available. We focused on nursing homes in this study as residents fitting the criteria for advanced dementia would most likely require nursing home level of care. In this study we did not involve healthcare assistants in core or wider meetings although their input was sought during assessments and they received training to improve EOL knowledge (Parks, 2005). The benefit of involving them in the core and wider meetings requires further investigation.”

Reviewer comment: Page 15 – it would be helpful if there was more clear linkage in this table to the key activities in Table 1 e.g. Core meetings – is that 1b? Table 2 – Individual reviews completed – what does this relate to in Table 1?

Response: We have added another column to Table 2 to link it more closely with Table 1.

Reviewer comment: Did each care home have a specified GP from one practice, or a lot of practices caring for different residents? Please clarify as this has implications for the commissioning of the Compassion Intervention in the future.

Response: This detail is included in Supplementary File 3 – context of each NH, but we have also added to the ‘context’ section of the results: “Both NHs were part of larger private companies and both had contracts with one GP surgery with the goal of having one GP oversee the medical care of all residents within the NH.”

Reviewer comment: Page 15 - In Table 1 it says that those delivering the social care (care

assistants?) were involved in component 1b but they are not mentioned in Table 2 as being present at core meetings– some discussion of why this was and the implications of this would be useful in the discussion section.

Response: HCAs were not involved in core meetings and we have clarified row 1b to include: “NH nursing staff responsible for resident’s needs” HCAs were consulted during the assessment process regarding any concerns or difficulties they experienced in caring for the resident. In the implications section we have also added: “Further work also needs to examine the applicability of the model to long term care settings where nursing care is not available. In this study we did not involve healthcare assistants in core or wider meetings although their input was sought during assessments and they received training which has been shown to improve EOL knowledge (Parks, 2005). The benefit of involving them in the core and wider meetings requires further investigation.”

Reviewer comment: Page 17 – information sessions for families – if sustainability was important would it have been better to train up the staff in this? - needs some reflection and discussion... Page 17 - Line 14 – what was the generated discussion about? This would be useful for readers to know as it would give some insight into families’ experiences – unless it is confidential

Response: Have added in the discussion: “There is the need for someone with the skills to discuss end of life with family carers and to provide staff training, given the high turnover of direct care staff in NHs (22). During family group sessions it was evident that carers had a poor understanding of dementia and wanted to learn about all aspects of dementia, not only about EOL. Staff in the facility lacked confidence in providing information to families and would require a considerable amount of development in EOL dementia care before a role of an ICL became redundant.”

Reviewer comment: Page 19 - Line 42 – unable to verbally communicate – still communicating in embodied ways

Response: Have added the word ‘verbally’

Reviewer comment: In the qualitative data reported there is an absence of any quotes from healthcare assistants – I would like to hear from them too to get a more in-depth understanding of the processes and outcomes from their perspective and in terms of the rigor of the analysis.

Response: We have added additional quotes from healthcare assistants and activity coordinators in the sections on Insight, Acceptance and Change.

Reviewer comment: Page 23 – the reason why NH wide outcomes were not easily obtained needs some explanation.

Response: We have revised to state: “NHs did not maintain electronic records of any of the NH-wide outcomes. Manual searches of daily logs and individual care plans were required. At NH1 resuscitation status was not documented consistently and at NH2 obtaining these data required reading of individual care plans. Due to these difficulties we reduced collection frequency to three time points (months 1, 4 and 7).”

Reviewer comment: Page 23 – Line 46-57 – Are these indicators of success? If so it would be helpful to mention them earlier in the paper.

Response: These outcome measures were reported in the Method section under “Qualitative and quantitative process data recorded by ICL”. We have added to these measures in the method section: “as possible indicators of quality of EOL care”

Reviewer comment: Page 24 – were some of those who died people who were eligible to take part in the study but whose families had not given consent? Were there other people who died who had not met the eligibility criteria? I think it would be useful to reflect on some of this in the discussion as it is relevant for the next stage of the Compassion Intervention.

Response: We have added in the discussion: “The criteria for inclusion may appear inappropriate given that none of the recruited residents died during the intervention period. However, three had died in the period between the ICL assessment and the research team trying to recruit the participant. In addition, another participant died a few weeks after the Intervention period ceased. The other deaths in the NHs were amongst residents who did not all have dementia. Also, there were residents who were eligible for the Intervention but who the ICL had not had time to assess during the Intervention period. Also, our larger cohort study [16], using similar eligibility criteria found that only 36% of residents with advanced dementia died during a nine month observation period, reflecting the difficulty in prognosing EOL in dementia. We advocate a proactive approach to addressing advance care planning and actively managing symptoms of pain and discomfort for all NH residents, with the need for particular attention to the unique needs of residents with advanced dementia and limited capacity to verbally communicate their needs.”

Reviewer comment: Page 25 Line 7/8 – carers mental health – I think this is the first mention of this I think this needs some clarification of where and why it was measured and the potential implications of it.

Response: The data from carers has been reported in both the method and results sections in sections titled ‘NH resident data’ and “Individual resident outcomes” with Table 4 presenting this data. We have revised the headings of these sections to make it clearer that it also includes carer data. We have added to existing text in the discussion the following: “Possibly distressed carers seeking support were more motivated to participate. Previous studies suggest that EOL discussions can improve carer satisfaction with EOL care (Engel, 2006). We have analysed practice relating to EOL conversations elsewhere (Saini, 2016).”

Reviewer comment: Page 25 - I think you need to present a much stronger argument based on the data you have collected to justify why you think your intervention provides a framework to optimise EOL care in accordance with the EAPC recommendations.

Response: While the intervention requires more field testing it does provide a framework for improving EOL care. To reduce the strength of the statement we have revised the statement to “Our Intervention provides a framework that may promote EOL care in accordance with EAPC recommendations. The Compassion Intervention supports many of the EAPC’s domains including: 2) person-centred care, communication and shared decision making; 3) setting care goals and advance planning; 6) avoiding overly aggressive, burdensome or futile treatment; 7) optimal treatment of symptoms and providing comfort; 8) psychosocial and spiritual support; 9) family care and involvement; and 10) education of the health care team.”

Reviewer comment: There needs to be more engagement with the challenges e.g. low recruitment of residents from among those eligible – what might have helped with this? What can be learned about recruitment from other care home studies or organisations such as ENRICH?

Response: In addition to other comments made in this response we have also added to the limitations: “The short time frame and the difficulty in scheduling weekly meetings to discuss assessments limited the number of residents who could be assessed and who were therefore eligible for recruitment for collecting individual outcome data. Often the person listed as a proxy decision

maker had minimal contact with the resident and felt unable to make decisions on their behalf, prohibiting recruitment of both carers and residents. Using professional consultees enabled involvement of isolated residents.”

Reviewer comment: Are weekly core meetings realistic?

Response: We agree that weekly meetings may be difficult in the NH setting. We had included in the discussion section: “Prior to working with this Intervention, NHs should consider the feasibility of weekly core meetings and how to incorporate assessments into existing processes”

Reviewer comment: I would like to see a much more explicit account of what the authors have learnt from doing this feasibility, what worked well and more detail on how the findings will inform future testing... More information about the cost-effectiveness of the intervention would be helpful.

Response: We have provided a much more detailed discussion to address this and other comments from Reviewer 1. In addition to other text added we have added: “We have information regarding the costs, time and skills required to engage an ICL. We also highlight the benefits of an ICL who was external to the NH to drive practice change and to provide independent support for family carers (21). For localities with good external multidisciplinary support for NHs, Compassion might be implemented by employing a full time ICL working across 2-3 NHs. However, for contexts such as NH2, external support from a range of disciplinary areas (not costed in this study) would require greater investment from commissioners. The extent to which the context of NH1 or NH2 reflects the typical level of support for UK NHs is unknown.”

Reviewer comment: Given that there were deaths in both care homes but not among those in the study, were the eligibility criteria for recruitment right? The criteria on page 5 for inclusion could describe many care homes residents who might live with that level of need for many months. What are the implications of this for how we define end of life care for people with dementia and what it entails and the staff competencies and staffing levels required?

Response: We have addressed this in the Discussion by adding: “The criteria for inclusion may appear inappropriate given that none of the recruited residents died during the intervention period. However, three had died in the period between the ICL assessment and the research team trying to recruit the participant. In addition, another participant died a few weeks after the Intervention period ceased. The other deaths in the NHs were amongst residents who did not all have dementia. Also, there were residents who were eligible for the Intervention but who the ICL had not had time to assess during the Intervention period. Also, our larger cohort study [16], using similar eligibility criteria found that only 36% of residents with advanced dementia died during a nine month observation period, reflecting the difficulty in prognosing EOL in dementia. We advocate a proactive approach to addressing advance care planning and actively managing symptoms of pain and discomfort for all NH residents, with the need for particular attention to the unique needs of residents with advanced dementia and limited capacity to verbally communicate their needs.”

Reviewer comment: Page 20 lines 52/53 touches on the need to change the culture from a task driven culture to a compassionate care culture... whether they think their approach is the right approach to changing cultures of care with reference to other literature.. e.g. (Froggatt, 2009... and McCormack, 2017)... Work has been done in care homes that shows that systems on their own are not enough to change practice but attention also needs to be paid to the lifeworld of the staff, not just in relation to training but also in terms of emotional support to deal with the nature of caring for very frail people at the end of life e.g. Froggatt, 2011...

Response: “The Compassion Intervention was underpinned by organisational change theory [18].

There has been few intervention studies developed in NHs in advanced dementia, but none that have used an external role such as an ICL to facilitate practice change. External facilitators of the education focused GSFCH, report concerns about their lack of time to enable adequate support [45]. The level of facilitation in the Compassion Intervention was higher than the ‘high facilitation’ reported in the GSFCH programme, and training on its own is unlikely to change resistant norms and practices [46]. The study using the most similar approach but has not been completed at the date of this paper may provide useful insights into the benefits of family case conferencing in the NH setting (Agar, 2015) with implementation of a similar role as ICL but from a nurse within the NH. This will provide a useful comparison for the importance of an internal or external ICL. Our implementation was flexible in responding to the unique needs of the different NH contexts and the holistic assessments undertaken by the ICL were crucial in providing insights to NH staff about gaps in existing care provision. The ICL implemented a relationship-centred approach which aimed to provide information and practical and emotional support to NH staff, family and residents (Nolan, 2004). However, other approaches to implementing practice change are also worth considering. For example, action research used in the NH setting has been useful in transforming task-driven approaches to approaches that engage staff more meaningfully with care processes to enable practice improvements (Lea, 2012).”

Reviewer comment: I’d like to see some discussion of whether the authors consider the assessment template to be comprehensive enough. Environmental and design issues such as excess noise and bright lights can be confusing for people with dementia and lead to agitation and poor quality of life. Were these factors considered as part of the intervention, or considered beyond its scope? If so it would be helpful to discuss them as a limitation with reference to the wider literature...

Response: We have added the following in the discussion: “The assessment template we developed aimed to be holistic covering a broad range of issues including the person’s physical, social, psychological and spiritual needs. Although observational assessments may have identified environmental factors that impacted on the resident’s wellbeing, these were not explicitly included in the assessment but could be important to include (Chaudhury, 2013). Further testing of the Intervention may lead to further refinement of the assessment and identify new elements over time. In addition, the assessment required some duplication of existing assessments undertaken in each NH. To address this issue we have added a checklist to prompt NHs to examine existing assessment domains rather than requiring another assessment template.”

VERSION 2 – REVIEW

REVIEWER	Julie Watson University of Edinburgh Scotland UK
REVIEW RETURNED	11-Apr-2017
GENERAL COMMENTS	Useful paper which clearly reports findings from a naturalistic feasibility study of implementing the Compassion Intervention and makes an important contribution to the field of knowledge.